# Impact of delayed sternal closure on wound infections following neonatal and infant cardiac surgery

Maria von Stumm[1], Yola Leps[2], Luca Jochheim[2], Victoria van Rüth[2], Urda Gottschalk[3], Goetz Mueller[3], Rainer Kozlik-Feldmann[3], Mark G. Hazekamp[4], Joerg S. Sachweh[2‡], Daniel Biermann[2‡]*

1 Department of Congenital and Pediatric Heart Surgery, German Heart Center Munich, Technical University of Munich, Munich, Germany, 2 Department of Congenital and Pediatric Heart Surgery, Children's Heart Clinic, University Heart & Vascular Center, University Medical Center Hamburg-Eppendorf, Hamburg, Germany, 3 Department of Pediatric Cardiology, Children's Heart Clinic, University Heart & Vascular Center, University Medical Center Hamburg-Eppendorf, Hamburg, Germany, 4 Department of Cardiothoracic Surgery, Leiden University Medical Center, Leiden, The Netherlands

☯ These authors contributed equally to this work.
‡ JSS and DB also contributed equally to this work.
* d.biermann@uke.de

**Data Availability Statement:** All relevant data are within the paper and its Supporting Information files.

## Abstract

### Objectives

Delayed sternal closure is a routine procedure to reduce hemodynamic and respiratory instability in pediatric patients following cardiac surgery, particularly in neonates and infants. In this setting, the possible links between sternal wound infection and delayed sternal closure are still a matter of debate. As a part of our routine, there was a low threshold for delayed sternal closure, so we reviewed our experience with sternal wound infections with a focus on potentially related perioperative risk factors, particularly delayed sternal closure.

### Methods

We retrospectively identified 358 operated neonates (37%) and infants (mean age 3.6 months) in our local congenital heart disease database between January 2013 and June 2017. Potential risk factors for sternal wound infections, such as age, gender, complexity (based on Aristotle- and STS-EACTS mortality category), reoperation, use of cardiopulmonary bypass, extracorporeal membrane oxygenation, mortality and delayed sternal closure (163/358, 46%), were subjected to uni- and multivariate analysis.

### Results

A total of 26/358 patients (7.3%) developed a superficial sternal wound infection. There were no deep sternal wound infections, no mediastinitis or sepsis. Applying univariate analysis, the prevalence of sternal wound infections was related to younger age, more complex surgery and delayed sternal closure. However, in multivariate analysis, sternal wound infection was only associated with delayed sternal closure (p = 0.013, odds ratio 8.6). Logistic

**Funding:** The authors received no specific funding for this work.

**Competing interests:** The authors have declared that no competing interests exist.

regression revealed the prevalence of delayed sternal closure to be related to younger age, complexity, and the use of extracorporeal membrane oxygenation.

## Conclusion

In patients younger than one year, sternal wound infections are clearly related to delayed sternal closure. However, in our cohort, all sternal wound infections were superficial and acceptable, considering the improved postoperative hemodynamic stability.

## Introduction

Delayed sternal closure (DSC) is a well-established procedure to reduce postoperative hemo-dynamic and respiratory instability in neonatal or infant patients following cardiac surgery. In other series, around 10% of pediatric cardiac patients leave the operating room with an open chest, and on average, the sternal closure is performed three days later [1–3]. The prevalence of sternal wound infections (SWI) following DSC ranges from 3.5% to 18.0% [1–13]. Apart from superficial wound infection following DSC, life-threatening complications including mediastinitis and sepsis have been described [1]. Furthermore, the length and costs of hospitalization are increased in cases of SWI [1, 14]. Despite all these negative implications, a consensus or guideline regarding indication for or timing of sternal closure in these patients is missing. Therefore, many aspects of DSC are still a matter of debate, including optimal preparation of pediatric cardiac patients before surgery (i.e., preoperative bathing with chlorhexidine-gluconate; timing and choice of preoperative antibiotic treatment), technical intraoperative aspects (i.e., use of foreign material for pericardial augmentation), and postoperative management (i.e., daily bathing with chlorhexidine-gluconate; duration of antibiotics) [13, 15].

Since we applied DSC liberally in hemodynamically unstable, complex cases, we sought to report our experience with DSC in neonates and infants following cardiac surgery, focusing on potentially related perioperative risk factors for SWI.

## Patients and methods

### Study design

Patients were retrospectively identified from the institutional congenital heart disease database. Inclusion criteria were surgery for congenital heart disease in patients <1 year of age with or without DSC between January 2013 and May 2017. All patients were assigned to STS-EACTS Congenital Heart Surgery Mortality Categories (STAT Mortality Categories) [16]. In addition, the Aristotle Score was calculated for each patient [17]. The primary study endpoint was the occurrence of SWI during an observation period of one year. SWI were classified as superficial or deep wound infections according to the guidelines from the Centers for Disease Control/National Healthcare Safety Network [18]. The criteria for superficial wound infections were given when cutis and subcutis were involved, and the infection occurred within 30 days after the operation. For definition of deep SWI, muscle and fascial layers had to be affected, and the infection occurred within 30 days after the procedure or within one year if foreign material was used at the index operation.

**Treatment of patients before and during primary surgery.** All patients were bathed one day before surgery. Cefuroxime (50mg/kg body weight) was routinely administered within

half an hour before surgical incision. Surgical washing prior to surgery was performed using Cutasept G (Bode Chemie, Hamburg, Germany), a skin antiseptic that comprises 2-propanol and benzalkoniumchloride, which was applied three times in succession. Subsequently, standard surgical draping with a foil dressing was used to seal the skin before incision. After the termination of cardiopulmonary bypass (CPB), another dose of cefuroxime was administered routinely. In the case of severe bleeding, a third dose was given according to the assessment of the anesthesiological team.

**Indication and technique of DSC.** The decision for DSC was made intraoperatively, based on clinical conditions like presence of cardiac edema, extent of low cardiac output, heart rhythm disturbances, or respiratory failure. Following stage one palliation for single-ventricle anatomy DSC was routinely performed. Also, the sternum was left open in patients requiring circulatory support by extracorporeal membrane oxygenation (ECMO) postoperatively.

Technically, the pericardium, sternum, and skin remained open after placing chest tubes and pacing wires following accurate hemostasis. In some cases, sterile gauze was left inside the mediastinum to optimize hemostasis. The skin incision was covered with a tailored 0.1 mm Gore-Tex surgical membrane (Gore-Tex, W. L. Gore & Associates, Flagstaff, USA) in all patients and sutured to the skin edges with a running 5–0 monofilament nonabsorbable suture. Next, the line where the synthetic patch was sutured to the skin was sealed hermetically with 10% iodine-povidone gel. Finally, a double layer of opSite* incise drape (Smith & Nephew, Auckland, New Zealand) was attached to the chest, covering the synthetic patch and stabilizing the cannulae in cases of ECMO.

DSC was routinely performed on the pediatric cardiac intensive care unit (PCICU). The main criteria for sternal closure were hemodynamic stability with only mild to moderate circulatory drug support, negative fluid balance, normal coagulation status, and lack of acidosis. The synthetic membrane was removed, and a mediastinal swab was routinely taken for microbiological analysis. Following surgical irrigation of the mediastinum with gentamycin/sodium chloride solution, the pericardium was substituted by a tailored Gore-Tex surgical membrane. Sternal closure was performed with absorbable 2–0 or 1–0 Vicryl® sutures (Ethicon, Norderstedt, Germany). Finally, the subcutaneous tissue was adapted with two layers of 4–0 absorbable sutures in a running fashion. For skin closure, intracutaneous running 5–0 Monocryl® (Ethicon) or single matching monofilament Donati sutures were used depending on skin integrity.

**Patient management during DSC.** All patients received weight-based intravenous ampicillin/sulbactam (150mg/kg body weight daily) until the time of the surgical procedure. For skin antisepsis on PCICU, Octeniderm® (Schuelke & Mayr, Norderstedt, Germany), which comprises octenidinedihydrochloride 0.1g, 1-propanol (ph.Eur.) 30g, 2-propanol (Ph.Eur.) 45g, was used. Ampicillin/sulbactam was continued at least until 48 hours after DSC. If the mediastinal fluid culture was positive, tailored antibiotic therapy was continued for 7 to 14 days. Patients were maintained on appropriate ventilatory and inotropic support until DSC. Sedation and nutrition management was individualized.

## Statistical analysis

Descriptive analysis is reported as mean, median, standard deviation, and minimum-maximum for continuous variables. For binary variables, the number of cases and relative frequencies were reported. Continuous variables were compared using Mann-Whitney-U-Test. Binary variables were compared using Fisher's exact test. Significant differences ($p < 0.05$) and tendencies towards significance ($p < 0.1$) in univariate testing were subjected to logistic regression analysis. In addition, the odds ratio was calculated, and the 95% confidence limits were

reported. All statistical analyses were performed using SPSS version 26 statistical package (IBM, Markham, Canada).

## Ethics approval

For the study, all authors obeyed the Declaration of Helsinki. The local ethics committee waived the requirement for informed consent since we are reporting a retrospective study of anonymized samples (ethics committee of the medical association Hamburg, Germany, approval number: WF-063/21).

## Results

### Demographic and outcome data

Over a 4.5-year period, 358 consecutive patients less than one year of age were operated via median sternotomy. Overall surgical mortality was 8.4% and was not related to SWI (p = 0.771). Potential risk factors for SWI, such as age, gender, complexity (based on Aristotle and STS–EACTS mortality category), reoperation, use of CPB, ECMO and DSC, were evaluated (Table 1).

**Infectious complications.** Overall, 26/358 (7.3%) patients developed a SWI. The prevalence of SWI was higher in DSC compared to primary sternal closure (PSC, 14.7% and 1%, respectively). All SWI were superficial and occurred within the first two weeks after surgery: 2–4 days after DSC and 2–3 days after PSC. No patient suffered from a deep SWI, mediastinitis or sepsis. In addition to the typical appearance, wound cultures were positive for pathogens in 10/26 patients with SWI, including Staphylococcus aureus (n = 4), Staphylococcus epidermidis (n = 4), and Escherichia coli (n = 2). All patients with SWI received antibiotic treatment based on the finding of the antibiogram. One infant in the DSC cohort required surgical debridement of the superficially infected and necrotic tissue and subsequent secondary skin closure.

The patients undergoing DSC were younger with more complex disease and clinical course (average STAT-Mortality Score 3.8 in the DSC group vs 2.1 in the PSC group). About a third (33%) of DSC patients had undergone ECMO circulatory support, whereas no patient in the PSC group received ECMO circulation support (Table 2).

**Table 1. Demographic and outcome data of 358 neonates and infants after surgery for congenital heart disease without or with SWI.**

| Variable | All cases (n = 358) | No SWI (n = 332, 92.7%) | SWI (n = 26, 7.3%) | Uni-variate sign. (p) | Multi-variate sign. (p) | Odds ratio |
|---|---|---|---|---|---|---|
| **Age at surgery (months)** | 3.7 ± 3.3 (3.6; 0.03–12) | 3.9 ± 3.3 (3.9; 0.03–12) | 1.2 ± 2.1 (0.3; 0.03–9.2) | 0.000 | 0.229 | |
| **Male gender (n)** | 209 (58.4%) | 195 (58.7%) | 14 (53.8%) | 0.682 | | |
| **Aristotle score** | 8.1 ± 2.5 (8; 3–15) | 8.0 ± 2.5 (8; 3–15) | 9.2 ± 2.0 (9.75; 6–14.5) | 0.001 | 0.513 | |
| **STAT mortality category** | 2.9 ± 1.3 (3; 1–5) | 2.8 ± 1.3 (3; 1–5) | 3.8 ± 0.6 (4; 2–5) | 0.007 | 0.440 | |
| **Redo (n)** | 66 (18.4%) | 64 (19.3%) | 2 (7.7%) | 0.191 | | |
| **CPB (n)** | 321 (89.7%) | 297 (89.5%) | 24 (92.3%) | 1.0 | | |
| **ECMO (n)** | 53 (14.8%) | 46 (13.9%) | 7 (26.9%) | 0.084 | 0.946 | |
| **DSC (n)** | 163 (45.5%) | 139 (41.9%) | 24 (92.3%) | 0.000 | 0.013 | 8.62 (1.6–47) |

Continuous values are presented as mean and standard deviation as well as median (minimum—maximum); Categorial values are presented as number (n) and relative frequency (%). SWI—sternal wound infection; DSC—delayed sternal closure; STAT Mortality Category–Mortality categories for congenital heart surgery from the Society of Thoracic Surgeons-European Association for Cardio-Thoracic Surgery [16]; ECMO—extracorporeal membrane oxygenation. CPB–cardiopulmonary bypass. Aristotle Score (Simple–to very complex. 1–15)

**Table 2. Demographic and outcome data of patients undergoing PSC vs. DSC.**

| Variable | PSC (n = 195, 54.5%) | DSC (n = 163, 45.5%) | Uni-variate sign. (p) | Multi-variate sign. (p) | Odds ratio |
|---|---|---|---|---|---|
| **Age at surgery (months)** | 5.4 ± 3.0 (5.5; 0.03–12) | 1.6 ± 2.2 (0.4; 0.03–9.9) | 0.000 | 0.002 | 0.990 (0.985–0.995) |
| **Male gender (n)** | 105 (53.4%) | 104 (63.8%) | 0.067 | 0.162 | |
| **Aristotle score** | 6.9 ± 1.8 (6; 3–11) | 9.5 ± 2.3 (10; 6–15) | 0.000 | 0.000 | 1.75 (1.42–2.16) |
| **STAT mortality category** | 2.1 ± 1.2 (2; 1–45) | 3.8 ± 0.8 (4; 1–5) | 0.000 | 0.000 | 2.09 (1.42–3.09) |
| **Redo (n)** | 49 (25.1%) | 17 (10.4%) | 0.000 | 0.053 | 0.384 (0.150–1.01) |
| **CPB (n)** | 175 (87.5%) | 146 (89.6%) | 1.0 | | |
| **ECMO (n)** | 0 | 53 (32.5%) | 0.000 | 0.000 | 56.75 (6.06–531) |
| **SWI (n)** | 2 (1%) | 24 (14.7%) | 0.000 | 0.043 | 7.85 (1.07–57.1) |

Continuous values are presented as mean and standard deviation as well as median (minimum-maximum); Categorical values are presented as number (n) and relative frequency (%). PSC—primary sternal closure; DSC—Delayed sternal closure; STAT Mortality Category–Mortality categories for congenital heart surgery from the Society of Thoracic Surgeons-European Association for Cardio-Thoracic Surgery [16]; ECMO—extracorporeal membrane oxygenation. CPB–cardiopulmonary bypass. Aristotle Score (Simple–to very complex. 1–15); SWI—sternal wound infection

## Risk factor analysis

Gender, the need for reoperation and the use of CPB were not associated with a higher rate of SWI. In univariate analysis the age at surgery, a higher Aristotle and STAT-Mortality Score and DSC were associated with SWI. Multivariate logistic regression analysis revealed DSC as the only variable related to SWI (p = 0.013, odds ratio of 8.6).

As mentioned above, 14.7% (n = 24) of the patients with DSC developed a SWI. In univariate analysis, only a younger age at surgery could be found to be a risk factor for SWI in patients with DSC. Interestingly, the timing to DSC, use of CPB or the STAT mortality score did not differ significantly between the children with and without SWI after DSC, as seen in Table 3.

## Discussion

Technical aspects of DSC and potential prevention strategies of surgical site infection are, particularly in young patients, still a matter of debate. The prevalence of SWI after DSC in our cohort was 14.7%, which is comparable to rates of other groups [1–13].

**Table 3. Comparison of the patients with and without SWI after DSC (n = 163).**

| Variable | No SWI (n = 139, 85.3%) | SWI (n = 24, 14.7%) | Uni-variate sign. (p) |
|---|---|---|---|
| **Age at surgery (months)** | 1.8 ± 2.4 (0.5; 0.03–9.9) | 0.8 ± 1.2 (0.3; 0.03–5.4) | 0.042 |
| **Male gender (n)** | 91 (65.5%) | 13 (54.2%) | 0.358 |
| **Aristotle score** | 9.5 ± 2.4 (10; 6–15) | 9.3 ± 2.0 (10; 6–14.5) | 0.805 |
| **STAT mortality category** | 3.8 ± 0.8 (4; 1–5) | 3.9 ± 0.6 (4; 3–5) | 0.781 |
| **Redo (n)** | 17 (12.2%) | 0 | 0.079 |
| **CPB (n)** | 124 (89.2%) | 22 (91.7%) | 1.0 |
| **ECMO (n)** | 45 (32.4%) | 7 (29.2%) | 0.817 |
| **Chest re-exploration (bleeding)** | 7 (5%) | 1 (4.2%) | 1.0 |
| **Time until DSC (days)** | 5.7 ± 6.1 (3; 1–32) | 4.8 ± 5.2 (3; 1–25) | 0.537 |

Continuous values are presented as mean and standard deviation as well as median (minimum-maximum); Categorical values are presented as number (n) and relative frequency (%). SWI—sternal wound infection; STAT Mortality Category–Mortality categories for congenital heart surgery from the Society of Thoracic Surgeons-European Association for Cardio-Thoracic Surgery [16]; ECMO—extracorporeal membrane oxygenation. CPB–cardiopulmonary bypass. Aristotle Score (Simple–to very complex. 1–15).

Currently, recommendations for the prevention of surgical site infection after median sternotomy in the pediatric population are limited, and an evidence-based guideline is lacking, mainly consisting of information gathered in retrospective single-center studies [2–12]. There is only one multicenter quality improvement study [13], including 4,198 children and one register-based study [1] analyzing data of 6,127 infants from the STS congenital heart surgery registry. The latter reported an overall infection rate following DSC of 18.7%, including sepsis in 8.2%, (superficial and deep) in 6.3%, and mediastinitis in 1.8% of patients. The operative complexity of cardiac surgery measured by STAT mortality categories was similar in the STS population when compared to our study cohort. However, infection rates following DSC in our sample tended to be lower [1]. Unfortunately, specific technical aspects of DSC were not described in detail by Nelson-McMillan and colleagues, limiting the comparability to some degree [1].

In 2019, Yabrodi et al. published impressive results following temporary skin and subcutaneous tissue closure above the open sternum instead of using a surgical membrane in pediatric cardiac patients with DSC [3]. In their cohort of 165 pediatric patients (age <1 year), final chest closure was achieved after an average period of 3 days, and the overall infection rate was 9% including—in addition to cases of sepsis and pneumonia—only one case of surgical site infection within the study period of five years. They speculated that recreating the natural skin barrier during DSC could be partly beneficial to minimize wound infections [3].

Additional improvements in the prevention of postoperative infections were reached by Woodward and colleagues [20]. They established a protocolized technical approach following pediatric cardiac surgery in a multicenter quality improvement project: daily chlorhexidine gluconate baths, individually assigned stethoscopes, door signage, sterile gel for echocardiographic studies while the sternum was open, wearing sterile gloves, caps, and masks, and controlling traffic into and out of patient's room during the actual closing procedure. Following these measures, infection rates were decreased from 5.7% in year one to 4.3% in year two [20].

## Antimicrobial treatment

There is currently no consensus on the optimal antimicrobial treatment strategy in pediatric patients with DSC. In our cohort, antibiotic therapy with ampicillin/sulbactam was continued until 48 hours after DSC in the absence of positive mediastinal fluid cultures. This two-day antibiotic treatment was shown to be not inferior to a five-day treatment regime following pediatric DSC in a non-inferiority trial by Philip and colleagues [15].

Furthermore, Hatachi et al. investigated the effect of different prophylactic antibiotic regimes on the occurrence of postoperative bloodstream and surgical site infections in pediatric DSC patients concluding that a broad-spectrum antibiotic therapy consisting of vancomycin and meropenem was more effective in preventing infection than the sole application of cefazolin [19].

## Potential risk factors for infective complications

Many potential risk factors for SWI following DSC are currently under consideration, including prematurity, the complexity of surgery (i.e., STAT mortality category 4–5), location of chest closure, perioperative usage of ECMO, and duration of DSC [1, 5, 20, 21]. In 2013, Harder et al. investigated risk factors for surgical site infections in a nested case-control study with 375 pediatric patients (age <18 years), who underwent congenital heart surgery and DSC [5]. They identified several risk factors for SWI, including duration of sternum left open, duration of employment of mediastinal chest tubes, duration of parenteral nutrition, duration of mechanical ventilation and length of hospital stay. Moreover, in the analysis of the STS

congenital heart surgery database, the rate of an infective complication rose to 35% after an open chest of seven days [1]. In addition, Harder et al. demonstrated that perioperative ECMO support was an independent predictor for SWI (odds ratio 2.92) [5]. However, in the analysis of the STS congenital heart surgery database, ECMO support was not determined as a relevant risk factor for SWI following DSC (odds ratio 0.77) [1]. In our analysis, none of these parameters were associated with an increased rate of SWI after DSC. Our patients with DSC were younger and sicker than the patients with PSC. The average STAT mortality was 3.8 in the DSC cohort, which can also have an impact on developing an SWI. However, when comparing the patients with and without SWI in the DSC group, only a younger age at surgery was a risk factor for SWI.

It is still unknown if the location of the chest closure (i.e. OR vs. PCICU) has an impact on infective complications. Though, in the analysis of Nelson-McMillan, chest closure in a non-OR setting was not associated with higher SWI rates [1]. In an analysis of Bowman and colleagues, children who underwent sternal closure in an open bay bed prone to higher traffic, distraction and missing laminar flow on PCICU showed a tendency towards higher infection rates when compared to their counterparts closed in a separate room with a door. In our cohort, sternal closure was performed at PCICU in low traffic areas with walls and a door. Interestingly, infection rates were slightly lower than the overall infection rates in Bowman's analysis for patients with closure in the OR and the non-OR setting (8.7% vs. 13.7%, respectively) [7].

## Negative clinical impact of an infective complication

In 2018, Alten and colleagues investigated the epidemiology and clinical impact of health-care-associated infections (HAI) on PCICU [21]. They retrospectively examined 9,356 medical and 11,485 surgical PCICU encounters within the Pediatric Cardiac Critical Care Consortium clinical registry [21]. In line with the previously discussed literature, the youngest patients (i.e. pre-term neonates) had the highest incidence of HAI when compared to other age groups (odds ratio 2.6; p<0.0001) [21]. Furthermore, mortality rates in surgical PCICU patients with HAI increased nearly tenfold in comparison to patients without infective complications (2.3% vs. 22.0%; p<0.0001) [21]. Unfortunately, it remains unclear if the increased mortality in patients with SWI was a consequence of the infective complication and/or due to cardiac failure.

Our and other series confirm that DSC after cardiac surgery in pediatric patients is associated with an increased rate of SWI. However, factors increasing the risk of SWI in the setting of DSC are not reported in unison. Although, younger age at the time of surgery seems to be a common and reproducible risk factor.

## Strengths and limitations

The strengths of our study are the large sample size and detailed patient information. The main limitation is the single-center analysis and its retrospective nature, limiting the analysis of certain factors and care processes potentially related to the risk of developing SWI during DSC, including the use of different materials for temporary chest closure, variations in the duration and extent of antibiotic regimes, and protocol-based measures for SWI prophylaxis. Therefore, conclusions drawn from the presented data should be confirmed by further prospective multi-institutional studies.

## Conclusion

DSC is a simple measure to reduce the postoperative risk for respiratory and hemodynamic instability following neonatal or infant cardiac surgery. However, it is associated with an

increased risk of SWI. Nevertheless, in our cohort, no deep SWI were documented. The occurrence of superficial infections could be a small price considering improved survival in critically ill patients. If DSC is necessary, thorough preoperative measures and postoperative care are important to avoid deep/severe SWI. The potential risk factors for the development of SWI after DSC are still under debate. Multi-institutional prospective studies are warranted to further analyze risk factors for postoperative SWI following DSC, as DSC itself remains unavoidable in many occasions.

## Supporting information

**S1 Dataset. Data set of all 358 patients enclosed in this study.** key—patient number for anonymization; DSC—delayed sternal closure (1—delayed sternal closure, 0—primary sternal closure); TUSC—time until DSC (days), Gender (0—female; 1—male); OR_age (age at index operation in days), ECMO—extracorporeal membrane oxygenation (1—ECMO; 0—no ECMO); CPB—cardiopulmonary bypass (1—CPB; 0—no CPB); Re_Ex (Chest Re-Exploration, 1—yes, 0—no, 99—no data available); ReOP—Redo (1—Reoperation, 0—no reoperation), STS-EACTS—STAT Mortality category, Mortality categories for congenital heart surgery from the Society of Thoracic Surgeons-European Association for Cardio-Thoracic Surgery; Aristotle—Aristotle Score—Aristotle Score (Simple–to very complex. 1–15); surg_mort—mortality (1—yes, 0—no).
(PDF)

## Author Contributions

**Conceptualization:** Joerg S. Sachweh, Daniel Biermann.

**Data curation:** Maria von Stumm, Yola Leps, Luca Jochheim, Victoria van Rüth.

**Formal analysis:** Yola Leps, Mark G. Hazekamp, Joerg S. Sachweh, Daniel Biermann.

**Investigation:** Maria von Stumm.

**Methodology:** Joerg S. Sachweh, Daniel Biermann.

**Project administration:** Joerg S. Sachweh, Daniel Biermann.

**Resources:** Rainer Kozlik-Feldmann.

**Supervision:** Goetz Mueller, Rainer Kozlik-Feldmann, Mark G. Hazekamp, Joerg S. Sachweh, Daniel Biermann.

**Validation:** Joerg S. Sachweh.

**Visualization:** Yola Leps, Joerg S. Sachweh, Daniel Biermann.

**Writing – original draft:** Maria von Stumm, Yola Leps.

**Writing – review & editing:** Yola Leps, Urda Gottschalk, Joerg S. Sachweh, Daniel Biermann.

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
