## [Decision Letter · Decision Letter 0]

31 Jan 2022

PONE-D-22-01068Impact of delayed sternal closure on wound infections following neonatal and infant cardiac surgeryPLOS ONE

Dear Dr. Biermann,

Thank you for submitting your manuscript to PLOS ONE. After careful consideration, we feel that it has merit but does not fully meet PLOS ONE’s publication criteria as it currently stands. Therefore, we invite you to submit a revised version of the manuscript that addresses the points raised during the review process.

 Please consider reviewer comments and revise your paper. Please submit point by point replies to comments and provide changes made.

We look forward to receiving your revised manuscript.

Kind regards,

Salil Deo

Academic Editor

PLOS ONE

Journal Requirements:

Additional Editor Comments:

Dear Author

Thank you for submitting your research to PLOS ONE

Please cionsider revising your manuscript based on the comments of 2 reviewers

Reviewers' comments:

Reviewer's Responses to Questions

**Comments to the Author**

1. Is the manuscript technically sound, and do the data support the conclusions?

Reviewer #1: Yes

Reviewer #2: Yes

2. Has the statistical analysis been performed appropriately and rigorously? 

Reviewer #1: Yes

Reviewer #2: Yes

3. Have the authors made all data underlying the findings in their manuscript fully available?

Reviewer #1: Yes

Reviewer #2: Yes

4. Is the manuscript presented in an intelligible fashion and written in standard English?

Reviewer #1: Yes

Reviewer #2: Yes

5. Review Comments to the Author

Reviewer #1: Thank you for the opportunity to review the manuscript on "Impact of Delayed Sternal Closure (DSC) of Wound Infection Following Neonatal and Infant Cardiac Surgery". DSC is commonly employed strategy after CPB in infants and DSC is well documented risk factor for superficial and deep wound infection. Authors have concluded the same although all their infections were superficial. Few comment:

1. In Introduction, line 53 please add "and" after " leave the operating room with an open chest" instead of comma; Also line 55 and 56 "mediastinitis" is technically a local SWI. I would change the "local SWI" to "superficial wound infection"

2. Authors have compared groups "with and without SWI" and "Primary vs Delayed sternal closure" but I feel it would be important to compare the patients in delayed sternal closure group (n=163) who had SWI (n=24)and who did not (n= 139), to identify risk factors i.e. time to close, any chest opened at bed side for bleeding or tamponade, duration of antibiotics etc.

I want to congratulate the authors on a nice manuscript.

Reviewer #2: This is a retrospective review of 358 operated neonates and infants who underwent repair of a variety of congenital heart defects and their sternum were closed in a delayed fashion. The authors looked at risk of sternal wound infection and evaluated its risk factors. The risk of sternal wound infection was 7.3% among the entire cohort and was related to delayed sternal closure. I have the following questions:

1. With open chest, the authors used ampicillin/sulbactam till the sternum is closed. This seems to be not as a broad spectrum as other combinations that are commonly used such as “Cefepime/Vancomycin” especially against virulent microorganisms such as pseudomonas. Please clarify the reason for that choice among other antibiotic combinations.

2. The use of Gentamycin for chest irrigation at the time of closure is not evidence-based, especially the Gentamycin has a narrow spectrum of coverage. Did the author evaluate if this decreased risk of wound infection or not? Any evidence to support its use?

3. Those on Extracorporeal membrane oxygenator support (ECMO), was there any change in their antibiotic coverage? Did they remain on antibiotics during the entire support period?

4. Any utilization for vacuum-assisted wound closure for any of these patients?

5. What practical measures the author will take or advise to further decrease risk of wound infection after pediatric cardiac surgery?

6. Have the authors evaluated other wound-related complications such as: breakdown in the incision, wound drainage without actual infection, and fat necrosis? We believe these complications are even more common than actual wound infection especially in neonates and small infants.

7. The authors should look into the relationship between the wound infection risk and the duration of open chest.

Thank you

6. PLOS authors have the option to publish the peer review history of their article (what does this mean?). If published, this will include your full peer review and any attached files.

Reviewer #1: No

Reviewer #2: **Yes: **Sameh M. Said, MBBCh, MD, FACC, FACS

---

## [Author Response · Author response to Decision Letter 0]

14 Mar 2022

Response to editors and reviewers:

Journal Requirements:

Comment 1: 

Please ensure that your manuscript meets PLOS ONE's style requirements, including those for file naming. The PLOS ONE style templates can be found at https://journals.plos.org/plosone/s/file?id=wjVg/PLOSOne_formatting_sample_main_body.pdf and https://journals.plos.org/plosone/s/file?id=ba62/PLOSOne_formatting_sample_title_authors_affiliations.pdf

Answer 1: 

We revised our manuscript and edited Title page, Manuscript body and file names.

Comment 2: 

Please provide additional details regarding participant consent. In the ethics statement in the Methods and online submission information, please ensure that you have specified (1) whether consent was informed and (2) what type you obtained (for instance, written or verbal, and if verbal, how it was documented and witnessed). If your study included minors, state whether you obtained consent from parents or guardians. If the need for consent was waived by the ethics committee, please include this information.

Answer 2: 

We edited our paragraph on ethical approval. We write now: For the study, all authors obeyed the Declaration of Helsinki. The local ethics committee waived the requirement for informed consent since we are reporting a retrospective study of anonymized samples (ethics committee of the medical association Hamburg, Germany, approval number: WF-063/21). A copy of the certificate is uploaded to the EditorialManager.

Comment 3: 

We note that you have indicated that data from this study are available upon request. PLOS only allows data to be available upon request if there are legal or ethical restrictions on sharing data publicly. For more information on unacceptable data access restrictions, please see http://journals.plos.org/plosone/s/data-availability#loc-unacceptable-data-access-restrictions.

Answer 3: 

There are no restrictions for uploading our data. We included our data set in the Supporting Information section. 

Comment 4:

Your ethics statement should only appear in the Methods section of your manuscript. If your ethics statement is written in any section besides the Methods, please delete it from any other section. 

Answer 4: 

We corrected this, and the ethics statement appears now in the Method section. 

 

Review Comments to the Author

Reviewer #1

Comment 1: 

In Introduction, line 53 please add "and" after " leave the operating room with an open chest" instead of comma; Also line 55 and 56 "mediastinitis" is technically a local SWI. I would change the "local SWI" to "superficial wound infection"

Answer 1: 

Thank you for the advise, we corrected these sentences. 

Comment 2: 

Authors have compared groups "with and without SWI" and "Primary vs Delayed sternal closure" but I feel it would be important to compare the patients in delayed sternal closure group (n=163) who had SWI (n=24) and who did not (n= 139), to identify risk factors i.e. time to close, any chest opened at bed side for bleeding or tamponade, duration of antibiotics etc.

Answer 2: 

Thank you for this suggestion. We compared those two groups you mentioned and added another table, as seen in the manuscript. The only significant risk factor in this analysis was a younger age at the time of surgery. 

Reviewer #2

Comment 1: 

With open chest, the authors used ampicillin/sulbactam till the sternum is closed. This seems to be not as a broad spectrum as other combinations that are commonly used such as “Cefepime/Vancomycin” especially against virulent microorganisms such as pseudomonas. Please clarify the reason for that choice among other antibiotic combinations.

Answer 1: 

Thank you for your comment. In the absence of positive cultures and with no signs of infection, we prefer using a broad-spectrum antibiotic treatment with minimal risk for adverse effects like kidney function impairment. Nonetheless, we extend therapy according to guidelines once the microbiological smears have been evaluated. We discuss this comment in terms of lacking evidence in the Discussion section of the manuscript.

Comment 2: 

The use of Gentamycin for chest irrigation at the time of closure is not evidence-based, especially the Gentamycin has a narrow spectrum of coverage. Did the author evaluate if this decreased risk of wound infection or not? Any evidence to support its use?

Answer 2: 

We totally agree with you. Currently, the evidence regarding antibiotic agents, dosages, application protocols and SWI definitions vary widely throughout studies. However, our standard-of-care-practice for wound closure based on the findings from a RCT from Fridberg et al. The researcher showed that local collagen-gentamicin reduced the risk for postoperative sternal wound infections in adults. (Friberg O, Svedjeholm R, Söderquist B, Granfeldt H, Vikerfors T, Källman J. Local gentamicin reduces sternal wound infections after cardiac surgery: a randomized controlled trial. Ann Thorac Surg. 2005 Jan;79(1):153-61; discussion 161-2. doi: 10.1016/j.athoracsur.2004.06.043. PMID: 15620935.) In everyday practice, we no longer use gentamycin rinsing solution in children and use only warm saline solution for cleaning the situs.

Comment 3: 

Those on Extracorporeal membrane oxygenator support (ECMO), was there any change in their antibiotic coverage? Did they remain on antibiotics during the entire support period?

Answer 3: 

Antibiotic treatment (ampicillin/sulbactam) was given throughout the open-chest phase and 48 hours thereafter. In patients on ECMO during the entire therapy period. Antibiotics were adjusted as necessary after review of the swabs.

Comment 4: 

Any utilization for vacuum-assisted wound closure for any of these patients?

Answer 4: 

We did not use any vacuum-assisted devices in small children (<1 year). In larger children, we have had some good experiences with vacuum dressing in recent years. This experience will be published separately in the future.

Comment 5: 

What practical measures the author will take or advise to further decrease risk of wound infection after pediatric cardiac surgery?

Answer 5: 

Thank you for this good question. As described in the manuscript, we think that our study should not claim to make general recommendations. We have significantly deepened the topic of sterility in PCICU in recent years and now use sterile gloves and sterile gel during ultrasound examinations. In addition, we have become more aggressive in treating superficial wound infections and defects directly surgically and additionally with antibiotics. This has meant that we have not had any mediastinitis or deep wound healing problems in the age group described.

Comment 6: 

Have the authors evaluated other wound-related complications such as: breakdown in the incision, wound drainage without actual infection, and fat necrosis? We believe these complications are even more common than actual wound infection especially in neonates and small infants.

Answer 6: 

Thank you for the question. In our cohort, wound-related complications such as suture dehiscence and wound drainage were summarized as superficial wound infection, if there were additional signs of infection including redness, swelling, warmer skin and / or fever. Mostly, it was wound dehiscence in the lower or upper wound pole leading to surgical revision in our experience. Here (in the last years) a reduction of the frequency was achieved by using SteriStrips. This seems to provide additional safety especially when newcomers to the profession perform sternal or skin closures. We have not observed fatty tissue necrosis in the age group under one year. However, the author team also thinks that this etiology is the most common cause of SWI in older children and adults.

Comment 7:

The authors should look into the relationship between the wound infection risk and the duration of open chest.

Answer 7: 

In our cohort, we were unable to show any correlation in this respect, which is probably due to statistical reasons. It could be because patients with very long open sternal wounds partly died before a wound-healing disorder became evident. However, we also think that this is an important aspect. We compared the periods of open chest in the group of patients with DCS and included it in the new table.

---

## [Decision Letter · Decision Letter 1]

20 Apr 2022

Impact of delayed sternal closure on wound infections following neonatal and infant cardiac surgery

PONE-D-22-01068R1

Dear Dr. Biermann,

We’re pleased to inform you that your manuscript has been judged scientifically suitable for publication and will be formally accepted for publication once it meets all outstanding technical requirements.

Kind regards,

Salil Deo

Academic Editor

PLOS ONE

Additional Editor Comments (optional):

Thank you very much for submitting your manuscript to PLOS One. We are delighted to let you know that your study has been accepted for publication in PLOS One.

Reviewers' comments:

Reviewer's Responses to Questions

**Comments to the Author**

Reviewer #2: All comments have been addressed

2. Is the manuscript technically sound, and do the data support the conclusions?

Reviewer #2: Yes

3. Has the statistical analysis been performed appropriately and rigorously? 

Reviewer #2: Yes

4. Have the authors made all data underlying the findings in their manuscript fully available?

Reviewer #2: Yes

5. Is the manuscript presented in an intelligible fashion and written in standard English?

Reviewer #2: Yes

6. Review Comments to the Author

Reviewer #2: The authors have responded to all the queries and answered all the questions and revised the manuscript accordingly

7. PLOS authors have the option to publish the peer review history of their article (what does this mean?). If published, this will include your full peer review and any attached files.

Reviewer #2: **Yes: **Sameh M. Said, MD

---

## [Editor Report · Acceptance letter]

11 May 2022

PONE-D-22-01068R1 

Impact of delayed sternal closure on wound infections following neonatal and infant cardiac surgery 

Dear Dr. Biermann:

I'm pleased to inform you that your manuscript has been deemed suitable for publication in PLOS ONE. Congratulations! Your manuscript is now with our production department. 

Kind regards, 

on behalf of

Dr. Salil Deo 

Academic Editor

PLOS ONE